# Clustering Data with Nonignorable Missingness using Semi-Parametric Mixture Models

**Marie Du Roy de Chaumaray** [1]   **Matthieu Marbac** [1]

## Abstract

We are concerned in clustering continuous data sets subject to nonignorable missingness. We perform clustering with a specific semi-parametric mixture, avoiding the component distributions and the missingness process to be specified, under the assumption of conditional independence given the component. Estimation is performed by maximizing an extension of smoothed likelihood allowing missingness. This optimization is achieved by a Majorization-Minimization algorithm. We illustrate the relevance of the approach by numerical experiments.

## 1. Introduction

Clustering is a useful tool to analyze large data sets because it aims to group the subjects into few homogeneous subpopulations. Mixture models permit to achieve the clustering purpose in a rigorous context (McLachlan & Peel, 2000; Chauveau et al., 2015) but the case where data have missingness is generally neglected. However, the data sets often contain missing values, like in social surveys. Thus, statistical analysis are performed on a complete data where missing values have been either removed or imputed. Removing subjects having missing values leads to severe bias and/or losses of efficiency (Molenberghs et al., 2008). Imputing missing values (Van Buuren, 2018) suffers from a lack of consistency because imputations are generally performed with a model different to the model used to cluster. Moreover, the missing not at random (MNAR) scenario (Little & Rubin, 2019), where the missingness depends on the missing values even conditional on the observed covariates, is often the case in practice (*e.g.,* higher-income respondents may decline to report income data) and the approaches mentioned above produce biased results in such a case. Statistical analysis,

under the MNAR scenario, generally requires the missingness process to be considered. However, few statistical methods permit this scenario because the models are often not identifiable based on the observed data.

Two clustering approaches allow data subject to the MNAR scenario to be analyzed. Thus, Chi et al. (2016) introduces the $K$-POD algorithm that extends the $K$-means to the case of missing data even if the missing mechanism is unknown. However, this approach suffers from the standard drawbacks of the $K$-means algorithm (*i.e.,* assumptions of spherical clusters and equals proportions of the clusters). Alternatively, using a *selection model* approach (see Little (1993) and the definition in Section 2), Miao et al. (2016) proposed a specific Gaussian mixtures and $t$-mixtures to analyze data under MNAR scenario. For such approach, the missingness process must be specified (probit and logit distributions are generally used). However, this approach produces strong bias if the parametric assumptions (made on the covariate distribution or on the missingness process) are violated.

In this paper, clustering is performed via a mixture model that uses a *pattern-mixture model* approach (see Little (1993) and the definition in Section 2) with non-parametric distributions. Thus, no assumptions are made on the data distribution or on the missingness process except that the variables are independent within components. Note that this assumption is quite standard for semi-parametric mixtures (Hall et al., 2003; Kasahara & Shimotsu, 2014; Chauveau et al., 2015; Zheng & Wu, 2019). For each mixture component, we estimate, for each variable, its probability to be observed and its conditional distribution given the variable is observed. We emphasize that our concern is clustering and not imputation or density estimation. Indeed, without adding assumptions, the distribution of the variables within component cannot be estimated by our procedure. Estimation of the semi-parametric mixture can be done by maximizing the smoothed likelihood (Levine et al., 2011). In this paper, we extend the concept of smoothed likelihood to mixed-type data. Indeed, the model implies continuous (the covariates) and binary (indicators of the missingness) variables. In our extension, only the distribution of the continuous variables are smoothed. Thus, the smoothed likelihood can be maximized by a Majorization-Minimization (MM) algorithm (Hunter & Lange, 2004).

---

[*]Equal contribution  [1]Univ. Rennes, Ensai, CNRS, CREST - UMR 9194, F-35000 Rennes, France. Correspondence to: Matthieu Marbac <matthieu.marbac-lourdelle@ensai.fr>.

*Presented at the first Workshop on the Art of Learning with Missing Values (Artemiss) hosted by the $37^{th}$ International Conference on Machine Learning (ICML).* Copyright 2020 by the author(s).

The paper is organized as follows. Section 2 introduces the semi-parametric mixture used for clustering data with nonignorable missingness. Section 3 presents the MM algorithm used for estimation. Section 4 illustrates the relevance of the approach on numerical experiments. Section 5 gives a conclusion.

## 2. Mixture for nonignorable missingness

### 2.1. The data

The observed sample is composed of $n$ independent and identically distributed subjects arisen from $K$ homogeneous subpopulations. Each subject is described by $d$ continuous variables and some realizations of these variables may be unobserved. The missingness process is allowed to be non-ignorable. Thus, the probability, for a variable, to be not observed is allowed to depend on the value of the variable itself and the subpopulation membership.

Each subject $i$ is described by a vector of three variables $(\boldsymbol{X}_i^\top, \boldsymbol{R}_i^\top, \boldsymbol{Z}_i^\top)^\top$ where $\boldsymbol{X}_i = (X_{i1}, \ldots, X_{id})^\top \in \mathbb{R}^d$ is set of continuous variables, $\boldsymbol{R}_i = (R_{i1}, \ldots, R_{id})^\top \in \{0,1\}^d$ indicates whether $X_{ij}$ is observed ($R_{ij} = 1$) and $\boldsymbol{Z}_i = (Z_{i1}, \ldots, Z_{iK})^\top$ indicates the subpopulation of subject $i$ ($Z_{ik} = 1$ if subject $i$ belongs to subpopulation $k$ and otherwise $Z_{ik} = 0$). Each subject belongs to one subpopulation such that $\sum_{k=1}^K Z_{ik} = 1$. The realizations of $\boldsymbol{Z}_i$ are unobserved and a part of the realizations of $\boldsymbol{X}_i$ can be unobserved too. Therefore, the observed variables for subject $i$ are $(\boldsymbol{X}_i^{\text{obs}\top}, \boldsymbol{R}_i^\top)^\top$ where $\boldsymbol{X}_i^{\text{obs}}$ is composed of the elements of $\boldsymbol{X}_i$ such that $R_{ij} = 1$ and the unobserved variables for subject $i$ are $(\boldsymbol{X}_i^{\text{miss}\top}, \boldsymbol{Z}_i^\top)^\top$ where $\boldsymbol{X}_i^{\text{miss}}$ is composed of the elements of $\boldsymbol{X}_i$ such that $R_{ij} = 0$.

### 2.2. General mixture model

We use mixture models in a purpose of clustering and not for density estimation. Clustering aims to estimate the subpopulation memberships given the observed variables (*i.e.,* the realization of $\boldsymbol{Z}_i$ given $(\boldsymbol{X}_i^{\text{obs}\top}, \boldsymbol{R}_i^\top)^\top$) without assumption on the missingness process (*i.e.,* no assumption are made on the conditional distribution of $\boldsymbol{R}_i \mid \boldsymbol{X}_i, \boldsymbol{Z}_i$). The probability distribution function (pdf) of $(\boldsymbol{X}_i^\top, \boldsymbol{R}_i^\top)^\top$ for subpopulation $k$ (*i.e.,* $Z_{ik} = 1$) is denoted by $g_k(\cdot)$. Thus, the pdf $(\boldsymbol{X}_i^\top, \boldsymbol{R}_i^\top)^\top$ is defined by the pdf of a $K$-component mixture

$$g(\boldsymbol{x}_i, \boldsymbol{r}_i) = \sum_{k=1}^K \pi_k g_k(\boldsymbol{x}_i, \boldsymbol{r}_i), \qquad (1)$$

where $\pi_k > 0$, $\sum_{k=1}^K \pi_k = 1$ and $g_k(\cdot; \boldsymbol{\theta})$ is pdf of component $k$. From (1), the distribution of the observed values can be defined by two approaches: the *selection model* and the *pattern-mixture model*. The approach named *selection model* defines the joint distribution of $(\boldsymbol{X}_i^\top, \boldsymbol{R}_i^\top)^\top \mid \boldsymbol{Z}_i$

as the product between the distribution of $\boldsymbol{X}_i \mid \boldsymbol{Z}_i$ and the distribution of $\boldsymbol{R}_i \mid \boldsymbol{Z}_i, \boldsymbol{X}_i$. This approach requires to model the missingness process (*i.e.,* the conditional distribution of $\boldsymbol{R}_i \mid \boldsymbol{Z}_i, \boldsymbol{X}_i$) and should be considered when the aim is to fit the marginal distribution of $\boldsymbol{X}_i$. Alternatively, the approach named *pattern-mixture model* defines the joint distribution of $(\boldsymbol{X}_i^\top, \boldsymbol{R}_i^\top)^\top \mid \boldsymbol{Z}_i$ as the product between the distribution of $\boldsymbol{R}_i \mid \boldsymbol{Z}_i$ and the distribution of $\boldsymbol{X}_i \mid \boldsymbol{Z}_i, \boldsymbol{R}_i$. Thus, using the *pattern-mixture model*, the pdf of component $k$ is given by

$$g_k(\boldsymbol{x}_i, \boldsymbol{r}_i) = g_k(\boldsymbol{r}_i) g_k(\boldsymbol{x}_i \mid \boldsymbol{r}_i). \qquad (2)$$

For clustering, the approach named *pattern-mixture model* should be preferred because it does not require to specify the missingness process, allows this process to be nonignorable and permits to easily obtain the conditional probabilities of the subpopulation membership given the distribution of the observed values

$$\mathbb{P}(Z_{ik} = 1 \mid \boldsymbol{x}_i^{\text{obs}}, \boldsymbol{r}_i) = \frac{g_k(\boldsymbol{x}_i^{\text{obs}}, \boldsymbol{r}_i)}{\sum_{\ell=1}^K \pi_\ell g_\ell(\boldsymbol{x}_i^{\text{obs}}, \boldsymbol{r}_i \boldsymbol{\theta})}. \qquad (3)$$

Indeed, integrating the pdf of component $k$ over the missing variables $\boldsymbol{X}_i^{\text{miss}}$, we have

$$g_k(\boldsymbol{x}_i^{\text{obs}}, \boldsymbol{r}_i) = g_k(\boldsymbol{r}_i) g_k(\boldsymbol{x}_i^{\text{obs}} \mid \boldsymbol{r}_i). \qquad (4)$$

Note that this approach does not permit to estimate the marginal distribution of $\boldsymbol{X}_i \mid \boldsymbol{Z}_i$ without adding assumptions on the missing process. Thus, the proposed approach can be used for clustering but not for density estimation.

### 2.3. Semi-parametric mixture for nonignorable missingness

A wide range of literature focuses on models assuming that conditionally on knowing the particular subpopulation the subject $i$ came from, its coordinates $\boldsymbol{X}_i$ are independent. Thus, we extend this model for nonignorable missingness. The couples of variables $(X_{ij}, R_{ij})^\top$ are assumed to be conditionally independent given $\boldsymbol{Z}_i$. Thus, the distribution of $\boldsymbol{R}_i \mid \boldsymbol{Z}_i$ is a product of Bernoulli distributions and the conditional density of $\boldsymbol{X}_i \mid \boldsymbol{Z}_i, \boldsymbol{R}_i$ is defined as the product of univariate densities. Thus, from (2), the pdf of component $k$ is also defined as

$$g_k(\boldsymbol{x}_i, \boldsymbol{r}_i) = g_k(\boldsymbol{r}_i; \boldsymbol{\tau}_k) \prod_{j=1}^d p_{kj}^{r_{ij}}(x_{ij}) q_{kj}^{1-r_{ij}}(x_{ij}), \quad (5)$$

with

$$g_k(\boldsymbol{r}_i; \boldsymbol{\tau}_k) = \prod_{j=1}^d \tau_{kj}^{r_{ij}} (1 - \tau_{kj})^{1-r_{ij}}, \qquad (6)$$

where $\boldsymbol{\tau}_k = (\tau_{k1}, \ldots, \tau_{kd})$, $\tau_{kj}$ is the probability that $X_{ij}$ is observed given that subject $i$ belongs to subpopulation

$k$, $p_{kj}(\cdot)$ is the conditional density of $X_{ij}$ given $Z_{ik} = 1$ and $R_{ij} = 1$ and $q_{kj}(\cdot)$ is the conditional density of $X_{ij}$ given $Z_{ik} = 1$ and $R_{ij} = 0$. Thus, clustering is achieved by modeling, for each subpopulation, the marginal probability of missingness and the conditional density given that the variable is observed. Integrating out the unobserved variables $\boldsymbol{X}_i^{\text{miss}}$, we have

$$g_k(\boldsymbol{x}_i^{\text{obs}}, \boldsymbol{r}_i; \boldsymbol{\theta}) = \sum_{k=1}^{K} \pi_k g_k(\boldsymbol{x}_i^{\text{obs}}, \boldsymbol{r}_i; \boldsymbol{\theta}), \qquad (7)$$

with

$$g_k(\boldsymbol{x}_i^{\text{obs}}, \boldsymbol{r}_i; \boldsymbol{\theta}) = g_k(\boldsymbol{r}_i; \boldsymbol{\tau}_k) \prod_{j=1}^{d} p_{kj}^{r_{ij}}(x_{ij}), \qquad (8)$$

where $\boldsymbol{\theta}$ groups all the finite parameters ($\pi_k$ and $\boldsymbol{\tau}_k$) and all the infinite parameters $p_{kj}(\cdot)$. Note, we do not need to estimate $q_{kj}(\cdot)$ for the clustering purpose but that this implies that we are not able to estimate the distribution of $\boldsymbol{X}_i \mid \boldsymbol{Z}_i$.

The following assumptions provide sufficient conditions for the model identifiability stated by Lemma 1 which is consequences of Allman et al. (2009).

**Assumption 1** *The $p_{kj}$'s are linearly independent, $\pi_k > 0$ and $\tau_{kj} > 0$.*

**Lemma 1** *If Assumption 1 holds true, then the model defined by (7)-(8) is identifiable, up to label swapping.*

# 3. Estimation by maximization of the smoothed likelihood

## 3.1. Maximum smoothed likelihood

To perform parameter estimation, we extend the approach of Levine et al. (2011) that uses the smoothed likelihood to the case of mixed-type variables. Indeed, the observed variables contains continuous variables $\boldsymbol{x}_i^{\text{obs}}$ and binary variables $\boldsymbol{r}_i$. Note that the smoothing is only performed on the densities and thus on the distributions of $\boldsymbol{x}_i^{\text{obs}}$. To perform parameter estimation, we consider that Assumptions 1-2 hold true.

**Assumption 2** *Let $\Omega_j$ a compact subset of $\mathbb{R}$ such that for $i = 1, \ldots, n$, $x_{ij} \in \Omega_j$, then we have $p_{kj} \in L_1(\Omega_j)$ and $\ln p_{kj} \in L_1(\Omega_j)$, for $j = 1, \ldots, d$.*

Let $S$ be the smoothing operator defined by

$$\mathcal{S}g_k(\boldsymbol{x}_i^{\text{obs}} \mid \boldsymbol{r}_i) = \prod_{j=1}^{d} (\mathcal{S}p_{kj}(x_{ij}))^{r_{ij}} \qquad (9)$$

and

$$\mathcal{S}p_{kj}(x_{ij}) = \int_{\Omega_j} \frac{1}{h} K\left(\frac{x_{ij} - u}{h}\right) p_{kj}(u) du, \qquad (10)$$

where $K$ is a kernel function and $h > 0$ its bandwidth. We consider the non linear smoothing operator defined by

$$\mathcal{N}g_k(\boldsymbol{x}_i^{\text{obs}}, \boldsymbol{r}_i; \boldsymbol{\theta}) = g_k(\boldsymbol{r}_i; \boldsymbol{\tau}_k) \exp\{\mathcal{S}\ln g_k(\boldsymbol{x}_i^{\text{obs}} \mid \boldsymbol{r}_i)\}.$$

where $g_k(\boldsymbol{x}_i^{\text{obs}} \mid \boldsymbol{r}_i) = \prod_{j=1}^{d} p_{kj}^{r_{ij}}(x_{ij})$.
The smoothed log-likelihood function is defined by

$$\ell_n(\boldsymbol{\theta}) = \sum_{i=1}^{n} \ln\left(\sum_{k=1}^{K} \pi_k \mathcal{N}g_k(\boldsymbol{x}_i^{\text{obs}}, \boldsymbol{r}_i; \boldsymbol{\theta})\right). \qquad (11)$$

Parameter estimation is performed by maximizing the smoothed likelihood over $\boldsymbol{\theta}$. This maximization is achieved by a MM algorithm presented in the next section.

## 3.2. Majorization-Minimization algorithm

The maximization on $\boldsymbol{\theta}$ of the smoothed log-likelihood function is performed via an MM algorithm. This iterative algorithm starts at the initial value of the parameters $\boldsymbol{\theta}^{[0]}$. At iteration $[r]$, it performs the following two steps

- Computing the smoothed probabilities of subpopulation memberships

$$t_{ik}(\boldsymbol{\theta}^{[r]}) = \frac{\pi_k^{[r]} \mathcal{N}g_k(\boldsymbol{x}_i^{\text{obs}}, \boldsymbol{r}_i; \boldsymbol{\theta}^{[r]})}{\sum_{\ell=1}^{K} \pi_\ell^{[r]} \mathcal{N}g_\ell(\boldsymbol{x}_i^{\text{obs}}, \boldsymbol{r}_i; \boldsymbol{\theta}^{[r]})}. \qquad (12)$$

- Updating the estimators

$$\pi_k^{[r+1]} = \frac{1}{n} \sum_{i=1}^{n} t_{ik}(\boldsymbol{\theta}^{[r]}), \qquad (13)$$

$$\tau_{kj}^{[r+1]} = \frac{\sum_{i=1}^{n} r_{ij} t_{ik}(\boldsymbol{\theta}^{[r]})}{\sum_{i=1}^{n} t_{ik}(\boldsymbol{\theta}^{[r]})} \qquad (14)$$

and

$$p_{kj}^{[r+1]}(u) = \frac{\sum_{i=1}^{n} r_{ij} t_{ik}(\boldsymbol{\theta}^{[r]}) \frac{1}{h} K\left(\frac{x_{ij} - u}{h}\right)}{\sum_{i=1}^{n} r_{ij} t_{ik}(\boldsymbol{\theta}^{[r]})}. \qquad (15)$$

The monotony of the algorithm is stated by Lemma 2 whose proof is similar to the proof of Theorem 1 in Levine et al. (2011). This implies that the algorithm converges to a local optimum of the smoothed log-likelihood, hence different random initializations should be performed.

**Lemma 2** *Let $\boldsymbol{\theta}^{[r]}$ and $\boldsymbol{\theta}^{[r+1]}$ be the estimators obtained at iterations $[r]$ and $[r + 1]$ respectively, we have $\ell_n(\boldsymbol{\theta}^{[r]}) \leq \ell_n(\boldsymbol{\theta}^{[r+1]})$.*

## 4. Numerical experiments

We generate complete data from a bi-component mixture with equal proportions and independence between variables within components such that $X_{ij} = (Z_{i1} - Z_{i2}) + \varepsilon_{ij}$ where the $\varepsilon_{ij}$ are independent from all the variables. Then, we add missing values from the logistic model

$$\mathbb{P}(R_{ij} = 0 \mid X_{ij}, \boldsymbol{Z}_i) =$$
$$(1 + \exp(\alpha(z_{i1} - z_{i2}) + \beta x_{ij}))^{-1}.$$

We consider data sets composed by $n = 100$ observations, $d = 4$ variables and three distributions for $\varepsilon_{ij}$ (standard Gaussian, Student with 3 degrees of freedom and exponential with rate 3). Parameters $\alpha$ and $\beta$ allow to set the impact of the subpopulation membership and the value of the variables on the missingness. For each scenario, we generated 100 data sets, we use a Gaussian kernel with bandwidth $h = n^{-1/5}$ and we compare the proposed method to the following approaches:

- *K-pod*: $K$-pod approach performed with the function *kpod* of the R package *kpodclustr* (Chi & Chi, 2014);

- *NPimputed*: non parametric mixture on the imputed data performed with the functions *np* and *imputePCA* of the R packages *mixtools* (Benaglia et al., 2009b) and *missMDA* (Josse & Husson, 2016).

To compare the methods, we compute the Adjusted Rand index (ARI; Hubert & Arabie (1985)) between the true partition and the estimators of the partition given by the methods. Table 1 presents the results. Results show that when the missing completly at random (MCAR) assumption holds true (*i.e.,* $\alpha = \beta = 0$) all the methods have the same performances. However, the proposed method outperforms the competing methods under the MNAR scenario.

We now compare the methods on three benchmark data presented in Table 2. Each data set contains a true partition which is not used during the estimation. We cluster the *original* data sets then we add missing values in the data sets by considering three scenarios: *MCAR* where each observation is missing with probability 0.2; missing at random (*MAR*) where variable $X_{2j-1}$ is always observed and variable $X_{2j}$ is not observed with probability $(1 + \exp(X_{2j-1}))^{-1}$; and *MNAR* where each variable $X_j$ is not observed with probability $(1 + \exp(X_j))^{-1}$.

For each scenario, we generated 10 data sets. Table 3 presents the ARI between the true partition and the estimated partitions. It confirms the results of the previous simulation. The three methods obtain similar results under the MCAR and MAR scenario. However, the proposed methods outperforms the competing methods under the MNAR scenario (except for the *coffee* data set where K-pod obtains slightly better results).

| $\varepsilon_{ij}$ | $(\alpha, \beta)$ | SPMNM | K-pod | NPimputed |
|---|---|---|---|---|
| Gaussian | (0,0) | 0.47 | **0.55** | 0.48 |
| | (1,0) | **0.74** | 0.31 | 0.39 |
| | (0,1) | **0.63** | 0.34 | 0.37 |
| | (1,1) | **0.85** | 0.11 | 0.22 |
| Student | (0,0) | 0.24 | 0.25 | **0.32** |
| | (1,0) | **0.59** | 0.12 | 0.21 |
| | (0,1) | **0.41** | 0.10 | 0.12 |
| | (1,1) | **0.74** | 0.02 | 0.07 |
| Exp | (0,0) | **0.87** | **0.87** | 0.86 |
| | (1,0) | **0.95** | 0.54 | 0.53 |
| | (0,1) | **0.96** | 0.68 | 0.65 |
| | (1,1) | **0.99** | 0.24 | 0.40 |

Table 1. Mean of the Adjusted Rand index obtained by the proposed semi-parametric mixture for nonignorable missingness (SPMNM), K-pod algorithm (K-pod) and non parametric mixture on imputed data (NPimputed). Best values are in bold.

| Name | $n$ | $d$ | $K$ | Reference |
|---|---|---|---|---|
| bank | 200 | 6 | 2 | Flury & Riedwyl (1988) |
| coffee | 43 | 12 | 2 | Streuli (1973) |
| wine | 178 | 13 | 3 | Forina & al (1991) |

Table 2. Information about the benchmark datasets.

| Data | Scenario | SPMNM | K-pod | NPimputed |
|---|---|---|---|---|
| Bank | original | **0.98** | 0.85 | **0.98** |
| | MCAR | **0.89** | 0.80 | 0.84 |
| | MAR | 0.57 | **0.64** | 0.63 |
| | MNAR | **0.71** | 0.55 | 0.58 |
| Coffee | original | **1.00** | **1.00** | **1.00** |
| | MCAR | 0.96 | 0.76 | **0.99** |
| | MAR | **1.00** | **1.00** | **1.00** |
| | MNAR | 0.87 | **0.92** | 0.80 |
| Wine | original | **0.95** | 0.90 | 0.91 |
| | MCAR | **0.83** | 0.79 | 0.80 |
| | MAR | **0.87** | 0.85 | 0.83 |
| | MNAR | **0.66** | 0.42 | 0.34 |

Table 3. Mean of the Adjusted Rand index obtained by the proposed semi-parametric mixture for nonignorable missingness (SPMNM), K-pod algorithm (K-pod) and non parametric mixture on imputed data (NPimputed). Best values are in bold.

## 5. Conclusion

The proposed method allows continuous data set with non-ignorable missingness to be clustered with no more assumption than the independence within components. Selecting the number of components is a difficult task that could be achieved by extending the approach of Kasahara & Shimotsu (2014) to the mixed-type data. Finally, a procedure of bandwidth selection should be investigated like in Benaglia et al. (2009a).

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
