# OpenReview forum: "Clustering Data with nonignorable Missingness using Semi-Parametric Mixture Models"
_ICML.cc/2020/Workshop/Artemiss — ICML Artemiss 2020_

### Official Review · AnonReviewer2 · 2020-06-17
**A clearly written paper which clearly fits the requirements for the Artemiss workshop**

**Rating:** 9
**Confidence:** 4

**Review:**

# Summary
The authors proposes a general clustering methods for continuous data with missing values. Their approach is based on semi parametric pattern-mixture models which allows to be agnostic both to the components distributions and the missingness process, preferring clustering to density estimation. Moreover, conditional independence of coordinate dimensions, given the component, is assumed. Inference is done by extending the smoothed likelihood approach of Levine et al. (2011) to the missing data framework. Sufficient conditions are given for identifiability.

# Impressions

__Pros__ :
- The authors' approach is described in a clear and sound way. The aim is to do clustering, which is clearly stated in the beginning of the paper, and the modelling choices are well justified with respect to that objective.
- The method is interesting for its generality both with respect to component distributions (semi-parametric approach) and the missingness scenario. Simulations show promising results for classical continuous distributions (even if the chosen parameters seems to generate well separated clusters), and also for the different missingness scenarii (MCAR, MAR and MNAR).
+ Real data experiments illustrates the interest of the method under MNAR scenario which is a crucial point of the paper.

__Cons__ :
It may be due to the lack of space (4 pages) but the experimental section is quite light w.r.t to:
- The distributions considered: all have finite variance, which fits the chosen Gaussian kernel. Maybe stable distributions could be considered (like Cauchy) and other kernels.
- The effect of dimensionality: it could be interesting to see the impact of $d$ on the estimation. Does the conditional independence of coordinate dimensions smoothen estimation problems that could arise in high-dimensional problems ?
- Time complexity of the MM algorithm could also be briefly discussed.

# Questions

1. Could you comment on the chosen Gaussian kernel bandwidth for the experiments $ h = n^{-1/5}$ ? Was it chosen via cross-validation or another hyper-parameter estimation technique ? It seems that it is, as in any kernel density estimation / smoothing problem, a critical parameter to set as stated in the conclusion.

2. Could you add a column on Table 1 and 3, specifying the average number of missing data in each cases ? when $\alpha = \beta = 0$ it is $50$\%, but I'm guessing it depends on the distribution, its parameter, and the values of $\alpha$ and $\beta$ ?

# Minor remarks
__Presentation suggestions__:
- Every important equation should be numbered, *e.g.* the parameter update formula.
- The acronym MNAR (missing non at random ?) should be defined once in the introduction
- Meaningless semantic in *MM algorihthm* (Hunter & Lange, 2004)): it's either Majorization-Minimization (when minimizing an objective) or Minorization-Maximization (when you maximize), but not Majorization-Minorization.  Since you use an MM algorithm for **maximisation**, shouldn't you use the latter ?


__Some typos__:
- Beginning of 2.1: "form" --> 'from'
- p.1, end of intro: missigness --> missingness
- abstract & introduction : Majorization-**Minorization** -> Majorization-**Minimization** (while Minorization-Maximization could be better here, cf. previous remark)
- p.3, before equation (3) : "Integrated out (...)" -> "Integrating out (...)"

---

### Official Review · AnonReviewer1 · 2020-06-23
**Well structured paper about clustering of continous data using semi parametric mixture models.**

**Confidence:** 3
**Rating:** 9

**Review:**

# SUMMARY:

The authors proposed an interesting method for the clustering of continous data. The semi parametric pattern-mixture model approach is used to defined the distribution of the observed values. The authors underline that the purpose of this method is clustering, thus no assumption about the missingness process  is needed. This method allows the missingness process to be nonignorable and to obtain the conditional probabilities of the sub population-membership. For the parameter estimation the approach of  the smoothed likelihood is used and to maximize the smoothed log-likelihood it it used the Majorization-Minimization algorithm.

# PROS:
- Perfectly in line with the topic of the conference.
- The assumptions and the goals of the model are explained very clearly.
- Originality of the proposed methodology.
- The article is well structured and well written.

# CONS:
- The authors should mention how the bandwidth of $h=n^{-1/5}$  of the Gaussian kernel has been defined.
- The acronyms MNAR, MAR and MCAR are not defined.
- It would have been interesting to give more space to the experiments and to the description of the dataset used.

## Final remark and suggestions
- For the MM algorithm, both in the "Abstract " and in the "1.Introduction" the authors refers to the Majorization-Minorization method while the paraghraph 3.2 is called  Majorization-Minimization algorithm.
- In the paragraph _"4. Numerical experiments"_ there are Table 1, Table 2 and Table 3. However, in the text Table 4 has been cited twice, probably instead of Table 1 and Table 3, because this table does not exist in the article.

---

### Decision · Program_Chairs · 2020-07-02

**Decision:**

Accept

**Comment:**

We're happy to accept this paper at Artemiss. We'll contact you soon to inform you about more details concerning the format of your presentation at the workshop, and the camera-ready version deadline. Please take into account the referee's comments to write the camera-ready version.